Sex- and stage-dependent expression patterns of odorant-binding and chemosensory protein genes in Spodoptera exempta

Dong Yonghao 1 2
Li Tong 3
Liu Jin 4
Sun Meixue 2
Chen Xingyu 2
Liu Yongjie lyj@sdau.edu.cn 1
Xu Pengjun xupengjun@163.com 2
1 College of Plant Protection, Shandong Agricultural University , Taian , Shandong Province , China
2 Qingdao Special Crops Research Center, Chinese Academy of Agricultural Sciences , Qingdao , Shandong Province , China
3 Institute of Plant Protection, Henan Academy of Agricultural Sciences , Zhengzhou , Henan Province , China
4 Shandong Agriculture and Engineering University , Jinan , Shandong Province , China
He Peng
Electronic publication date: 2021 Sep 13
Publication date: 2021
Volume: 9
Electronic Location ID: e12132
Received 2021 Apr 1; Accepted 2021 Aug 18
Copyright: ©2021 Dong et al.
Copyright year: 2021
Copyright holder: Dong et al.
License: This is an open access article distributed under the terms of the Creative Commons Attribution License, which permits unrestricted use, distribution, reproduction and adaptation in any medium and for any purpose provided that it is properly attributed. For attribution, the original author(s), title, publication source (PeerJ) and either DOI or URL of the article must be cited.
License URL: https://creativecommons.org/licenses/by/4.0/

Keywords: Spodoptera exempta, Odorant-binding protein, Chemosensory protein, Transcriptome, Gene expression

Funding: Agricultural Science and Technology Innovation Program ASTIP-TRIC04 Natural Science Foundation of Shandong Province ZR2020QC127 Distinguished Young Scholars from Henan Academy of Agricultural Sciences 2020JQ05 This study were funded by the Agricultural Science and Technology Innovation Program (Grant No. ASTIP-TRIC04), the Natural Science Foundation of Shandong Province (Grant No. ZR2020QC127) and the Fund for Distinguished Young Scholars from Henan Academy of Agricultural Sciences (Grant no. 2020JQ05). The funders had no role in study design, data collection and analysis, decision to publish, or preparation of the manuscript.

==============================
As potential molecular targets for developing novel pest management strategies, odorant-binding proteins (OBPs) and chemosensory proteins (CSPs) have been considered to initiate odor recognition in insects. Herein, we investigated the OBPs and CSPs in a major global crop pest (Spodoptera exempta). Using transcriptome analysis, we identified 40 OBPs and 33 CSPs in S. exempta, among which 35 OBPs and 29 CSPs had intact open reading frames. Sequence alignment indicated that 30 OBPs and 23 CSPs completely contained the conserved cysteines. OBPs of lepidopteran insects usually belonged to classical, minus-C, and plus-C groups. However, phylogenetic analyses indicated that we only identified 28 classical and seven minus-C OBPs in S. exempta, suggesting that we might have missed some typical OBPs in lepidopteran insects, probably due to their low expression levels. All of the CSPs from S. exempta clustered with the orthologs of other moths. The identification and expression of the OBPs and CSPs were well studied in insect adults by transcriptional analyses, and herein we used samples at different stages to determine the expression of OBPs and CSPs in S. exempta. Interestingly, our data indicated that several OBPs and CSPs were especially or more highly expressed in larvae or pupae than other stages, including three exclusively (SexeOBP13, SexeOBP16 and SexeCSP23) and six more highly (SexeOBP15, SexeOBP37, SexeCSP4, SexeCSP8, SexeCSP19, and SexeCSP33) expressed in larvae, two exclusively (SexeCSP6 and SexeCSP20) and three more highly (SexeOBP18, SexeCSP17, and SexeCSP26) expressed in pupae. Usually, OBPs and CSPs had both male- and female-biased expression patterns in adult antennae. However, our whole-body data indicated that all highly expressed OBPs and CSPs in adults were male-biased or did not differ, suggesting diverse OBP and CSP functions in insect adults. Besides identifying OBPs and CSPs as well as their expression patterns, these results provide a molecular basis to facilitate functional studies of OBPs and CSPs for exploring novel management strategies to control S. exempta.

Introduction

Insects have evolved diverse olfactory systems, which play pivotal roles in their survival in terms of foraging, mating, predator evasion, and reproduction (Van Naters & Carlson, 2006; Leal, 2013). Accordingly, insect behavior can be significantly affected by chemical signals during the process of olfactory perception, which provides novel opportunities to develop pest management strategies (e.g., push-pull strategy) (Cook, Khan & Pickett, 2007; Qiao et al., 2009; Sun et al., 2012; Tunstall & Warr, 2012). During the initiation of olfactory perception, the odorant binding proteins (OBPs) and chemosensory proteins (CSPs), which represent two gene families, are considered to regulate the first step of odor recognition (Pelosi et al., 2006; Zhou, 2010; Leal, 2013; Brito, Moreira & Melo, 2016). Therefore, OBPs and CSPs are considered potential molecular targets for developing novel pest management strategies.

Low identities of amino acid sequences among OBPs within or between species suggest that these genes evolve rapidly (Vieira & Rozas, 2011; Campanini & De Brito, 2016). However, there are two conserved structural features of OBPs: the hydrophobic cavity from at least six α-helical domains and the conserved cysteine (Cys) residues (Lagarde et al., 2011; Pelosi et al., 2014; Spinelli et al., 2012; Leal, Nikonova & Peng, 1999). Based on the number of Cys residues, insect OBPs are divided into five groups: classical OBPs (six conserved Cys), plus-C OBPs (eight conserved Cys), minus-C OBPs (four conserved Cys), dimer OBPs (12 conserved Cys), and atypical OBPs (9–10 conserved Cys) (Zhou, 2010). Generally, most insect OBP genes are highly expressed in the antennae (Cui et al., 2017; Li et al., 2017; Tang et al., 2019; Zhao et al., 2018). However, some OBP genes were expressed in other tissues, which also showed a similar odor molecule-binding function (Sun et al., 2017; Sun et al., 2019).

Like the OBPs, the amino acid sequences of CSPs also show low identities within or between species but have four conserved Cys residues forming two disulfide bridges (Leal, Nikonova & Peng, 1999; Honson & Plettner, 2006; Pelosi et al., 2006). In contrast to highly expressed OBPs in chemosensory tissues, the CSPs are broadly expressed in both olfactory and non-olfactory tissues, suggesting additional functions besides olfactory recognition (He et al., 2017; Sun et al., 2015; Wang et al., 2017; Zhang et al., 2013; Zhang et al., 2017). For example, the CSPs located in olfactory tissues were shown to execute an olfactory function, while a CSP in Locusta migratoria was found to be involved in behavioral phase changes (Ozaki et al., 2005; Guo et al., 2011).

Species from the genus Spodoptera are well-known as major crop pests worldwide (Early et al., 2018; Pogue, 2002; Rose, Dewhurst & Page, 2000; Zhang et al., 2017; Li et al., 2021). The African armyworm, S. exempta, is one of the most important migratory crop pests of cereals in sub-Saharan Africa, attacking maize, millet, rice, and wheat, among others (Rose, Dewhurst & Page, 2000). To develop novel strategies for controlling this pest, we focused on identifying genes in the OBP and CSP families in S. exempta using transcriptome analysis. We also showed the expression pattern of these genes using transcriptome data at different life stages, including the larvae on the first day of the fifth instar stage, pupae, and adult males and females.

Materials & Methods

Identification of OBP and CSP genes by transcriptome analysis in S. exempta

Individuals of S. exempta used in this study were from the colonies established with individuals collected in South Africa in 2014 and Tanzania in 2017, which the larvae were reared using standard artificial diet and adult moths were provided with 5% sugar water at 27 °C with a 14:10, light:dark photoperiod (Xu et al., 2020). RNA-seq were performed with whole body of individuals at different stages, including the 1st day of fifth instar stage larvae, pupae, males and female as described previously (Xu et al., 2020). To find OBPs and CSPs in S. exempta, 151 OBP protein sequences from 75 insect species (Vogt, Grosse-Wilde & Zhou, 2015) and 152 CSP protein sequences from 10 insect species (Walker et al., 2019; Vieira & Rozas, 2011) were used as subjects in local blastx searches. The unigenes uncovered in the transcriptome datasets of S. exempta (Table S1) (Xu et al., 2020) were used as queries. An e-value threshold of 1 × 10−5 was used in these searches. The putative OBP and CSP genes were further filtered by performing conserved domain searches in the NCBI conserved domain database using an e-value threshold of 1 × 10−2. Genes that did not contain a conserved PBP_GOBP domain (Accession number pfam01395) for the OBP genes and a conserved OS-D domain (Accession number pfam03392) for the CSP genes were excluded from further analyses.

Identification of the sequences of OBP and CSP genes by PCR and Sanger sequencing

According to the sequences showing high identities with OBPs and CSPs from other insects, we also designed primers to determine these sequences with PCR and Sanger (Table S2). Using TRIzol (Invitrogen, Grand Island, USA), total RNA was extracted from the whole bodies of S. exempta at different stages, including larvae, pupae, male adults and female adults. cDNA was synthesized with TranScript Reverse Transcriptase (Transgen, Beijing, China). The PCR program was as follows: 30 s at 94 °C, 30 s at 55 °C, and 30 s at 72 °C, for 40 cycles.

Motif analysis of OBPs and CSPs

The protein motifs of the S. exempta OBPs and CSPs were discovered using the MEME (version 5.3.3) (Bailey et al., 2009) online server (https://meme-suite.org/meme/tools/meme). A zoops (Zero or One Occurrence per Sequence) distribution pattern was employed, and five motifs were obtained with a minimum width of six and a maximum width of 10. The results were visualized using TBtools (Chen et al., 2000).

Sequence and phylogenetic analysis

The open reading frames (ORFs) of the OBP and CSP genes were predicted using the ORF Finder Tool at NCBI (http://www.ncbi.nlm.nih.gov/gorf/gorf.html) using the standard genetic code. The putative N-terminal signal peptides were predicted using SignalP V4.0 (Petersen, Brunak & Heijne, 2011). The molecular mass and isoelectric point of the predicted proteins were batch-computed in the sequence manipulation suite (Stothard, 2000). The 194 and 78 lepidopteran OBP and CSP protein sequences were retrieved and used to uncover the phylogenetic positions of the S. exempta OBP and CSP genes, respectively. Herein, the S. exempta OBP and CSP genes involved in the phylogenetic analysis were those containing intact ORF regions. The sequences were aligned using MUSCLE, as implemented in MEGA 7.0 (Kumar, Stecher & Tamura, 2016). The phylogenetic analysis was performed using IQ-TREE 1.6.6 (Nguyen et al., 2015). The substitution model was selected using ModelFinder (Kalyaanamoorthy et al., 2017) with the Bayesian information criterion. The ultrafast bootstraps were resampled at 5,000 runs to assess the support for each node. The phylogenetic trees were visualized using the ggtree R package (Yu et al., 2017).

Expression analysis of the OBP and CSP genes

Previously, we collected samples of S. exempta from single gender mixed pairs, including the larvae on the first day of the fifth instar, pupae, and adult males and females, and performed RNA-seq on these samples (Xu et al., 2020). The RNA-Seq data were submitted to the NCBI Sequence Read Archive (SRA) database (Table S1). For the gene expression analysis, the number of expressed tags was calculated and then normalized to transcripts per million tags (TPM) using RSEM software packages (Li & Dewey, 2011). The expression of the identified OBPs and CSPs genes was calculated based on the TPM values from the transcriptome data of S. exempta, which had five or six replicates at each point, and included samples from larvae, pupae, and adult males and females. To confirm the results of RNA-seq, samples at different stages, including larvae, pupae, adult males and females, were collected and using β-actin and GAPDH as reference genes, qRT-PCR with sybgreen method was performed in 20 µl reaction agent comprised of 1 µl of template DNA, 2 ×Premix Ex Taq (Takara), 0.2 µM of each primer, using a 7500 Fast Real-time PCR System (Applied Biosystems) (Table S2). Thermal cycling conditions were: 40 cycles of 95 °C for 5 s, 60 °C for 34 s. The samples of each group were biologically replicated three times. Statistical analyses were conducted using Graphpad InStat 3. A one-way ANOVA with a Tukey test at 0.05 significant level were used to determine the significance of the expression levels of the OBP and CSP genes at different life stages.

Results

Identification of the OBP and CSP genes in S. exempta

By functional annotation, a total of 40 OBP genes (named SexeOBP1-40) were identified in S. exempta from the transcriptome pool (Xu et al., 2020), of which 35 OBP sequences contained complete ORFs (Table S3, Supplemental Informations 12, 13). Twenty-nine of the 35 OBPs with intact ORFs had a signal peptide at their N-terminal (Table S3). The sequence alignment showed that 24 OBPs had six conserved Cys residues with members in the classical OBP group (SexeOBP1–2, 4–9, 11, 13–14, 16–17, 20–24, 27, 31–33, and 35–36), and six OBPs had four conserved Cys with members in the Minus-C OBP group (SexeOBP3, 12, 18–19, 30, and 34) (Fig. 1, Supplemental Informations 13, 14). The other five OBPs did not have the conserved four or six Cys, but according to the conserved C2 and C5, four of these belonged to the classical OBP group (SexeOBP15, 25–26, and 28) and one belonged to the minus-C OBP group (SexeOBP10) (Fig. 1, Supplemental Informations 13, 14). A total of 33 CSPs were identified in S. exempta, of which 29 CSPs (SexeCSP1–29) had intact ORFs and 25 CSPs (SexeCSP1–8, 10–12, 14–23, and 25–28) had a signal peptide at their N-terminal (Table S2, Supplemental Informations 15, 16). The amino acid sequence alignment of the CSPs containing complete ORFs indicated that 23 CSPs (SexeCSP1, 3–8, 11–12, 14–16, and 18–28) had the four conserved Cys (Fig. 2, Supplemental Informations 16, 17). By PCR and Sanger, 31 OBPs and 25 CSPs were successfully amplified and sequenced, which containing all the highly expressed genes (TPM >10, Table S2).

Figure 1 Amino acid sequence alignment of odorant-binding proteins (OBPs) with intact open reading frames in Spodoptera exempta.

Yellow boxes show the conserved cysteines. The members of minus-c OBP family are highlighted.

Figure 2 Amino acid sequence alignment of chemosensory proteins (CSPs) with intact open reading frames in Spodoptera exempta.

Yellow boxes show the conserved cysteines.

Analysis of motif patterns of OBPs and CSPs in S. exempta

Most of the 24 classical OBPs with six conserved Cys covered the C-pattern of lepidopteran OBPs “C1-X25-30-C2-X3-C3-X36-42-C4-X8-14-C5-X8-C6” (Xu et al., 2009), except for SexeOBP6–8, 23, and 27 (Table S3). The six minus-C OBPs also fit the lepidopteran C-pattern, aside from lacking C2 and C5 (Table S5). All of the 23 CSPs with four conserved Cys fit the C-pattern of lepidopteran CSPs “C1-X6-C2-X18-C3-X2-C4” (Xu et al., 2009) (Fig. 2). To further study the characteristic region of the OBP and CSP proteins with intact ORFs in S. exempta, the motifs of these proteins were analyzed using MEME. The results indicated that there were five motifs in the OBPs and CSPs of S. exempta (Figs. 3 and 4). For the 30 OBPs with six or four conserved Cys, nine (SexeOBP12, 18–20, 24, 31, 33–34, and 26) had the 3-1-2 motif pattern, eight (SexeOBP7–8, 13–14, 16–17, 21, and 27) had the 3-1-4-2 motif pattern, seven (SexeOBP3, 6, 9, 23, 30, 32, and 35) had the 1-2 motif pattern, five (SexeOBP1–2, 4, 11, and 22) had the 3-1-5-2 motif pattern, and one (SexeOBP5) had the 1-4-2 motif pattern (Fig. 3). For the 23 CSPs with four conserved Cys, 13 (SexeCSP1, 3, 5, 7–8, 12, 15, 19–20, 23, 25–26, and 28) had the 3-1-2 motif pattern, four (SexeCSP6, 11, 16, and 18) had the 3-1 motif pattern, three (SexeCSP4, 21–22) had the 3-1-2-4 motif pattern, two (SexeCSP14 and 27) had the 3-5 motif pattern, and one (SexeCSP24) had the 3-2 motif pattern (Fig. 4).

Figure 3 The distribution pattern and SeqLogo of the protein motifs in Spodoptera exempta odorant-binding proteins (OBPs).

The x-axis of motif distribution pattern indicates the length of OBP proteins. The SeqLogo of motifs are visualized by TBtools.

Figure 4 The distribution pattern and SeqLogo of the protein motifs in Spodoptera exempta chemosensory proteins (CSPs).

The x-axis of motif distribution pattern indicates the length of CSP proteins. The SeqLogo of motifs are visualized by TBtools.

Phylogenetic analysis of OBPs and CSPs

In the OBP gene phylogenetic tree (Fig. 5), SexeOBP3, 12, 18–19, 30, and 34 were grouped into the minus-C OBP clade, which agreed with their sequence features. The other 28 S. exempta OBPs were scattered on the phylogenetic tree; however, none were grouped into the plus-C or PBP/GOBP clades. In the CSP gene phylogenetic tree (Fig. 6), all S. exempta CSPs clustered with the lepidopteran classical CSPs (except for SexeCSP2, which lacks the four conserved Cys).

Figure 5 Maximum likelihood tree of lepidopteran odorant-binding proteins (OBPs).

The protein names and sequences of the OBPs used here are listed in Supplemental Information 13 and the reference (Gu et al., 2015). In total, 228 OBPs are used, including 35 Spodoptera exempta OBPs, 43 Bombyx mori OBPs, 38 Spodoptera litura OBPs, 36 Spodoptera littoralis OBPs, 26 Helicoverpa armigera OBPs, 17 Spodoptera exigua OBPs, and 33 Agrotis ipsilon OBPs.

Figure 6 Maximum likelihood tree of lepidopteran chemosensory proteins (CSPs).

The protein names and sequences of the CSPs used here are listed in Supplemental Information 16 and the reference (Li et al., 2020). In total, 107 CSPs are used, including 29 Spodoptera exempta CSPs, 15 Plutella xylostella CSPs, 13 Papilio xuthus CSPs, 6 Helicoverpa armigera CSPs, 20 Cnaphalocrocis medinalis CSPs, and 24 Bombyx mori CSPs.

Sex- and stage-dependent expression patterns of OBP and CSP genes in S. exempta

The number of unigenes specifically expressed (TPM value > 1) in larvae, pupae, adult males, and adult females were 6,301, 8,901, 10,112, and 12,068, respectively (Fig. 7A). The principal component analysis with the unigene expression data clearly distinguished the life stages and sex of individuals (Fig. 7B). To identify the differentially expressed genes (DEGs) at different stages, we screened the DEGs using RSEM under the conditions of padj < 0.05 and —log2(foldchange)— > 1, from which the number of DEGs identified were 11,588, 9,441, 10,069, and 8,109 in larvae, pupae, males, and females, respectively (Supplemental Information 18). We performed a pathway enrichment analysis on the DEGs. Interestingly, the top three pathways that were significantly enriched were the same in the four groups: protein digestion and absorption, neuroactive ligand–receptor interaction, and pancreatic secretion pathways (Fig. S1).

Figure 7 The gene expression patterns among the developmental stages in Spodoptera exempta.

(A) Principal component analysis (PCA) analysis of gene expression at different life stages. The gene expression matrix among the samples are used in PCA, and then visualized by stats R package. (B) Venn diagram showing the number of genes expressed at different life stages.

The expression levels of the OBPs and CSPs were shown with the TPM values. For the OBPs, 11 (SexeOBP9, 13, 15–16, 18, 26, 29–31, 35, and 37) of 40 were relatively highly expressed (TPM value > 10), among which two (SexeOBP13 and 16) were specifically expressed in larvae (genes with an expression level greater than 1 TPM), three (SexeOBP29, 31, and 35) were specifically expressed in adult males, two [(SexeOBP15 (P < 0.0001, d.f. = 3, 19, F = 36.551) and SexeOBP37 (P < 0.0001, d.f. = 3, 19, F = 91.700)] were more highly expressed in larvae than in other stages, one [SexeOBP18 (P < 0.0001, d.f. = 3, 19, F = 29.163)] was more highly expressed in the pupae than in other stages, and three [SexeOBP9 (P < 0.0001, d.f. = 3, 19, F = 35.333), SexeOBP26 (P < 0.0001, d.f. = 3, 19, F = 110.87), and SexeOBP30 (P < 0.0001, d.f. = 3, 19, F = 64.471)] were more highly expressed in adult males than in other stages (Fig. 8, Table S7). The genes SexeOBP2 and SexeOBP12 were specifically expressed with relatively low levels in adult males (TPM values < 10) (Table S7). There were no OBPs specifically or more highly expressed in adult females and all of the expressed OBPs in adults had significantly higher expression levels in males than in females [SexeOBP4 (P = 0.0120, d.f. = 10, t = 3.060), SexeOBP8 (P < 0.0001, d.f. = 10, t = 7.315), SexeOBP9 (P = 0.0007, d.f. = 10, t = 4.816), SexeOBP11 (P = 0.0076, d.f. = 10, t = 3.334), SexeOBP15 (P = 0.0002, d.f. = 10, t = 5.616), SexeOBP18 (P < 0.0001, d.f. = 10, t = 22.818), SexeOBP24 (P = 0.0147, d.f. = 10, t = 2.944), SexeOBP26 (P = 0.0004, d.f. = 10, t = 5.214), and SexeOBP30 (P < 0.0001 d.f. = 10, t = 11.685)], except for SexeOBP20 (P = 0.5641, d.f. = 10, t = 0.5965) and SexeOBP37 (P = 0.3329, d.f. = 10, t = 1.018) (Fig. 8, Table S7).

Figure 8 Expression patterns of odorant-binding proteins (OBPs) among different developmental stages in Spodoptera exempta based on the transcripts per million tags (TPM) values.

The TPM values were normalized by the logarithmic scale with base 2, and then scaled by row. The color and size of the circle indicates the gene expression level. Darker colors and larger circles indicate the genes were highly expressed in the samples. The heatmap was visualized by TBtools.

For the CSPs, 16 (SexeCSP2, 4–5, 8–10, 12–13, 17, 19, 23, 25–26, 28, 30, and 33) of 33 were relatively highly expressed (TPM value >10), of which only one CSP (SexeCSP23) was specifically expressed in larvae (Fig. 9, Table S8). One of the 16 CSPs with high expression levels was expressed with no significant differences among the different life stages [SexeCSP10 (P = 0.058, d.f. = 3, 19, F = 2.960)], while the high expression of the other 14 CSPs significantly differed according to the different life stages, e.g., four were highly expressed in adult males [SexeCSP2 (P < 0.0001, d.f. = 3, 19, F = 95.165), SexeCSP5 (P < 0.0001, d.f. = 3, 19, F = 18.880), SexeCSP25 (P < 0.0001, d.f. = 3, 19, F = 211.83), and SexeCSP30 (P < 0.0001, d.f. = 3, 19, F = 47.486)], four in larvae [SexeCSP4 (P < 0.0001, d.f. = 3, 19, F = 34.952), SexeCSP8 (P < 0.0001, d.f. = 3, 19, F = 19.168), SexeCSP19 (P = 0.0058, d.f. = 3, 19, F = 5.722), and SexeCSP33 (P < 0.0001, d.f. = 3, 19, F = 23.152)], two in pupae [SexeCSP17 (P < 0.0001, d.f. = 3, 19, F = 27.019) and SexeCSP26 (P < 0.0001, d.f. = 3, 19, F = 43.573)], and the others in two or three stages [SexeCSP9 (P < 0.0001, d.f. = 3, 19, F = 53.073) in larvae and males, SexeCSP12 (P < 0.0001, d.f. = 3, 19, F = 38.825) in pupae and males, SexeCSP13 (P = 0.0005, d.f. = 3, 19, F = 9.399) in larvae, males, and females, and SexeCSP28 (P = 0.0321, d.f. = 3, 19, F = 3.618) in larvae, pupae, and males] (Fig. 9, Table S8). Ten of the 16 OBPs were expressed at significantly higher levels in males than in females [SexeCSP2 (P < 0.0001, d.f. = 10, t = 6.892), SexeCSP5 (P = 0.0007, d.f. = 10, t = 4.872), SexeCSP9 (P < 0.0001, d.f. = 10, t = 6.744), SexeCSP12 (P = 0.0005, d.f. = 10, t = 5.065), SexeCSP17 (P = 0.0014, d.f. = 10, t = 4.377), SexeCSP19 (P = 0.0015, d.f. = 10, t = 4.319), SexeCSP25 (P < 0.0001, d.f. = 10, t = 11.322), SexeCSP26 (P = 0.0051, d.f. = 10, t = 3.568), SexeCSP28 (P = 0.0012, d.f. = 10, t = 4.452), and SexeCSP30 (P = 0.0389, d.f. = 10, t = 2.376)] (Fig. 9, Table S7). The genes SexeCSP6 and SexeCSP20 were specifically expressed in pupae with low expression levels (Fig. 9, Table S8).

Figure 9 Expression patterns of chemosensory proteins (CSPs) among the different developmental stages in Spodoptera exempta based on the transcripts per million tags (TPM) values.

The TPM values were normalized by a logarithmic scale with base 2 and then scaled by row. The color and size of a circle indicates the gene expression level. Darker colors and larger circles indicate the genes were highly expressed in the samples. The heatmap was visualized by TBtools.

The highly expressed genes of OBPs and CSPs (TPM > 10) in S. exempta were validated with sybgreen qPCR using β-actin and GAPDH as reference genes (Table S2). The results were consistent with the data from RNA-seq except for SexeOBP9, 30, 37, and SexeCSP5, 9, 12, 19, 30 (Figs. S2, S3, Supplemental Information 19).

Discussion

The genes in the OBP and CSP families are highly diverse with low identities, and the number of OBPs and CSPs significantly differed among insect species due to gene duplication and loss (Vieira & Rozas, 2011). Based on genome sequence data, the number of OBP and CSP genes showed significant differences among different insect orders. For example, there were 41–86 OBPs but no more than eight CSPs in the Diptera, of which the mosquitoes had more OBPs and CSPs than other flies (He et al., 2016; Vieira & Rozas, 2011; Venthur & Zhou, 2018), while 32–51 OBPs and approximately 20 CSPs were annotated in Lepidoptera and Coleoptera (Tribolium castaneum) (Gong et al., 2007; Gong et al., 2009; Vieira & Rozas, 2011; Vogt, Grosse-Wilde & Zhou, 2015; Vizueta et al., 2020). However, the numbers of OBPs and CSPs in insects from the Hymenoptera, Hemiptera, and Phthiraptera were less, with 21 OBPs and six CSPs in Apis mellifera (Hymenoptera), 15 OBPs and 13 CSPs in Acyrthosiphon pisum (Hemiptera), and 10 OBPs and nine CSPs in Myzus persicae (Hemiptera) (Vieira & Rozas, 2011; Wang et al., 2019; Zhou et al., 2010).

Recently, transcriptome analyses have been widely used to identify OBPs and CSPs due to the low expense of such analyses, although the annotated genes might be less than those identified with genome data due to pseudogenes and genes with very low expression levels (Vogt, Grosse-Wilde & Zhou, 2015; Xu et al., 2009; Venthur & Zhou, 2018; Zhang et al., 2018). In the current study, 40 OBPs (including 35 OBPs with intact ORFs) and 33 CSPs (including 29 CSPs with intact ORFs) were identified by transcriptome analysis of first-day fifth instar larvae, pupae, and adult males and females of S. exempta, similar to other species within the genus Spodoptera (e.g., S. littoralis (45 OBPs and 22 CSPs) and S. frugiperda (36 OBPs and 21 CSPs)) (Legeai et al., 2014; Walker et al., 2019). Typical OBPs and CSPs usually contain a signal peptide at their N-terminal; however, many OBPs and CSPs from the transcriptome data had no signal peptide, and one of the OBPs that was without a signal peptide showed normal function in M. persicae (Gu et al., 2015; Wang et al., 2021; Zhu et al., 2020). Our results indicated that seven of 35 OBPs and four of 29 CSPs did not contain the signal peptide. Additionally, five OBPs and six CSPs had intact ORFs but the number of conserved Cys was less than four. Therefore, we cannot fully conclude that we acquired the complete coding sequences of these genes due to a mismatch in the sequence assembly.

According to number of conserved Cys numbers, the OBPs were divided into five groups, of which the atypical OBPs have only been found in mosquitoes (Zhou et al., 2010; He et al., 2016). Based on the genome and transcriptome data, lepidopterans had the three most common types of OBPs, among which the number of classical OBPs > minus-C OBPs > plus-C OBPs, and only one dimer OBP was annotated in Danaus plexippus (Gu et al., 2015; Vogt, Grosse-Wilde & Zhou, 2015). However, we only found 24 classical and six minus-C OBPs in S. exempta. Consistent with this, the tree-based analysis indicated that only classical and minus-C OBPs were found in S. exempta. However, the transcriptome data suggested there were PBP/GOBP, classical, minus-C, and plus-C OBPs in two other species from the same genus (S. litura and S. littoralis) (Gu et al., 2015; Walker et al., 2019). The absence of PBP/GOBP and Plus-C OBPs in S. exempta might be due to low expression levels of these OBPs in our samples. Lepidopteran OBPs contained a conserved C-pattern in the form of “C1-X25-30-C2-X3-C3-X36-42-C4-X8-14-C5-X8-C6” (Xu et al., 2009). Interestingly, five of 30 OBPs in S. exempta showed different C-patterns [SexeOBP6 (C3-X45-C4-X15-C5), SexeOBP23 (C1-X19-C2), and SexeOBP7, SexeOBP8, and SexeOBP27 (C3-X43-C4)], which expanded the known C-patterns of lepidopteran OBPs. Generally, there were higher identities among insect CSPs than OBPs. Indeed, our results indicated that all 23 CSPs fit the conserved motif “C1-X6-C2-X18-C3-X2-C4” (Xu et al., 2009).

Expression analyses have been used to investigate the functions of OBPs and CSPs; e.g., the PBPs for detecting the sex pheromones of S. littoralis were more highly expressed in males than in females (Gu et al., 2015). Previously, most expression patterns of OBPs and CSPs have been investigated in adults (Cui et al., 2017; Gu et al., 2015; Walker et al., 2019; Wang et al., 2021). For the first time, we annotated and investigated the expression patterns at different life stages in S. exempta. Interestingly, there were two OBPs specifically expressed in larvae, and two OBPs in larvae and one in pupae had higher expression levels than at the other stages. The tissue-specific analysis suggested that most OBPs were more highly expressed in the antennae, and few were expressed in other tissues including the leg, brain, and body (Cui et al., 2017; Gu et al., 2015; Sun et al., 2017; Walker et al., 2019; Wang et al., 2021). In S. littoralis and S. litura, the OBPs with similarly tissue-specific expression patterns clustered together (Gu et al., 2015; Walker et al., 2019). According to the above references and the phylogenetic tree, nine OBPs (SexeOBP2, 4, 9, 11, 20, 22, 25–26, and 28) and four OBPs (SexeOBP23, 30, 32, and 35) were more highly expressed in the antennae and body respectively, and eleven OBPs (SexeOBP5, 7–8, 13–14, 18, 24, 27, 31–32, and 34) were expressed without a tissue-specific pattern (Gu et al., 2015; Walker et al., 2019). The expression pattern in the antennae indicated there were both male- and female-specific expression patterns in S. littoralis and S. litura (Gu et al., 2015; Walker et al., 2019). However, using the transcriptional data generated from whole-body samples, our data indicated that there were no female-biased OBPs and most of the expressed OBPs in adults were either expressed explicitly in males or had higher levels in males than in females.

The CSPs are usually expressed broadly in olfactory and non-olfactory tissues due to their relation to chemosensory and non-chemosensory processes (He et al., 2017; Sun et al., 2015; Walker et al., 2019; Wang et al., 2017). Like the OBPs, most CSPs have been previously studied in adults only. Our data indicated that SexeCSP23 was expressed explicitly at high levels in larvae, and two CSPs (SexeCSP6 and SexeCSP20) were specifically expressed at low levels in pupae. Unlike both the male and female-biased expression patterns of CSPs in S. littoralis (Walker et al., 2019), the CSPs with high expression levels were either male-biased (10 SexeCSPs) or were not differentially expressed between the sexes (six SexeCSPs). These results suggest that both the OBPs and CSPs might play essential roles outside of the antennae in adults, as well as in larvae and pupae. The function of these OBPs and CSPs should be investigated with gene knockout tools and bioassay, e.g., CRISPR/Cas9 system, by which PBP1/PBP3 were proved to play important roles in detecting sex pheromones in Chilo suppressalis (Dong et al., 2019) and OBP83a/OBP83b play roles in deactivation og odorant responses in Drosophila (Scheuermann & Smith, 2019).

Conclusions

In conclusion, we identified 40 OBPs (named SexeOBP1–40) and 33 CSPs (named SexeCSP1–33) in S. exempta by transcriptome analysis, of which 35 OBPs and 29 CSPs had intact ORFs. The sequence alignment analysis indicated that 30 OBPs and 23 CSPs completely contained the conserved Cys. The tree-based analyses indicated that 28 SexeOBPs clustered with classical OBPs and seven SexeOBPs clustered with minus-C OBPs, suggesting that we may have missed some OBPs that are typical of lepidopterans, possibly due to their low expression levels in S. exempta. The transcriptional analyses indicated that several OBPs and CSPs were specifically or more highly expressed in larvae or pupae than other life stages, including three specifically and six more highly expressed genes in larvae, and two specifically and three more highly expressed genes in pupae. The OBPs and CSPs had both male- and female-biased expression patterns in the antenna of the moths; however, our data from whole-body samples indicated that all of the highly expressed OBPs and CSPs in adults were either male-biased or did not differ between the sexes, suggesting diverse functions of OBPs and CSPs in insect adults.

Supplemental Information

Supplemental Information 1 The top 10 enrichment KEGG pathways in the DEGs uncovered in different developmental stages in S. exempta

Click here for additional data file.

Supplemental Information 2 Detection of highly expressed genes encoding odorant-binding proteins with sybgreen qPCR in S. exempta using β-actin and GAPDH as reference genes

(a) SexeOBP 9 (F = 0.555, d.f. =3,8, P = 0.659); (b) SexeOBP 13 (F = 72.571, d.f. =3,8, P = 0.000); (c) SexeOBP 15 (F = 9.771, d.f. =3,8, P = 0.005); (d) SexeOBP 16 (F = 16.651, d.f. =3,8, P = 0.001); (e) SexeOBP 18 (F = 42.290, d.f. =3,8, P = 0.000); (f) SexeOBP 26 (F = 76.012, d.f. =3,8, P = 0.000); (g) SexeOBP 29 (F = 34.62, d.f. =3,8, P = 0.000); (h) SexeOBP 30 (F = 28.574, d.f. =3,8, P = 0.000); (i) SexeOBP 31 (F = 107.519, d.f. =3,8, P = 0.000); (j) SexeOBP 35 (F = 459.605, d.f. =3,8, P = 0.000); (k) SexeOBP 37 (F = 152.93, d.f. =3,8, P = 0.000). Mean ± SE. Different letters showed significant difference (one-way ANOVA).

Click here for additional data file.

Supplemental Information 3 Detection of highly expressed genes encoding chemosensory proteins with sybgreen qPCR in S. exempta using β-actin and GAPDH as reference genes

(a) SexeCSP 2 (F = 1037.806, d.f. =3,8, P = 0.000); (b) SexeCSP 4 (F = 523.254, d.f. =3,8, P = 0.000); (c) SexeCSP 5 (F = 147.502, d.f. =3,8, P = 0.000); (d) SexeOBP 8 (F = 663.712, d.f. =3,8, P = 0.000); (e) SexeCSP 9 (F = 179.16, d.f. =3,8, P = 0.000); (f) SexeCSP 10 (F = 145.377, d.f. =3,8, P = 0.000); (g) SexeCSP 12 (F = 180.618, d.f. =3,8, P = 0.000); (h) SexeCSP 13 (F = 359.123, d.f. =3,8, P = 0.000); (i) SexeCSP 17 (F = 3131.788, d.f. =3,8, P = 0.000); (j) SexeCSP 19 (F = 759.292, d.f. =3,8, P = 0.000); (k) SexeCSP 23 (F = 587.196, d.f. =3,8, P = 0.000); (l) SexeCSP 25 (F = 31.917, d.f. =3,8, P = 0.000); (m) SexeCSP 26 (F = 1734.3, d.f. =3,8, P = 0.000); (n) SexeCSP 28 (F = 537.055, d.f. =3,8, P = 0.000); (o) SexeCSP 30 (F = 580.952, d.f. =3,8, P = 0.000); (p) SexeCSP 33 (F = 1583.71, d.f. =3,8, P = 0.000). Mean ± SE. Different letters showed significant difference (one-way ANOVA).

Click here for additional data file.

Supplemental Information 4 Data output for each pool of S. exempta

Click here for additional data file.

Supplemental Information 5 Primers used in this study

Click here for additional data file.

Supplemental Information 6 The OBPs identified in S. exempta by transcriptome

Click here for additional data file.

Supplemental Information 7 The CSPs identified in S. exempta by transcriptome

Click here for additional data file.

Supplemental Information 8 Conserved C-Pattern in OBPs of S. exempta

Click here for additional data file.

Supplemental Information 9 GenBank accession numbers of insect CSP genes used in the phylogenetic

Click here for additional data file.

Supplemental Information 10 The TPM values of S. exempta OBPs in different samples

L = larvae, P = pupae, M = male adults, F = female adults.

Click here for additional data file.

Supplemental Information 11 The TPM values of S. exempta CSPs in different samples

L = larvae, P = pupae, M = male adults, F = female adults.

Click here for additional data file.

Supplemental Information 12 The nucleotide sequences of OBPs in S. exempta

Click here for additional data file.

Supplemental Information 13 The amino acid sequences of OBPs in S. exempta

Click here for additional data file.

Supplemental Information 14 Amino acid sequence alignment of odorant-binding proteins (OBPs) with intact open reading frames in S. exempta

Click here for additional data file.

Supplemental Information 15 The nucleotide sequences of CSPs in S. exempta

Click here for additional data file.

Supplemental Information 16 The amino acid sequences of CSPs in S. exempta

Click here for additional data file.

Supplemental Information 17 Amino acid sequence alignment of chemosensory proteins (CSPs) with intact open reading frames in S. exempta

Click here for additional data file.

Supplemental Information 18 The DEGs at different stages in S. exempta

Click here for additional data file.

Supplemental Information 19 The Ct value of OBPs and CSPs with qPCR

Click here for additional data file.

Additional Information and Declarations

Competing Interests

Author Contributions

Data Availability

The authors declare there are no competing interests.

Yonghao Dong and Tong Li performed the experiments, analyzed the data, prepared figures and/or tables, authored or reviewed drafts of the paper, and approved the final draft.

Jin Liu analyzed the data, prepared figures and/or tables, authored or reviewed drafts of the paper, and approved the final draft.

Meixue Sun performed the experiments, prepared figures and/or tables, authored or reviewed drafts of the paper, and approved the final draft.

Xingyu Chen performed the experiments, authored or reviewed drafts of the paper, and approved the final draft.

Yongjie Liu conceived and designed the experiments, analyzed the data, authored or reviewed drafts of the paper, and approved the final draft.

Pengjun Xu conceived and designed the experiments, performed the experiments, analyzed the data, prepared figures and/or tables, authored or reviewed drafts of the paper, and approved the final draft.

The following information was supplied regarding data availability:

The raw data are available in the Supplementary Files. The RNA-Seq data are available at the NCBI Sequence Read Archive (SRA) database (SRR8594173 –SRR8594180).

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
