# Peer review of "Sex- and stage-dependent expression patterns of odorant-binding and chemosensory protein genes in Spodoptera exempta"

_PeerJ, doi:10.7717/peerj.12132_

## Round 0.1 · original submission · Major Revisions

Please revise your manuscript thoroughly according to the reviewers comments.

[Reviewer 1 ·

Basic reporting

no comment

Experimental design

no comment

Validity of the findings

no comment

Additional comments

In the MS, authors identified odorant-binding proteins (OBPs) and chemosensory proteins (CSPs) in and established the expression patterns in Spodoptera exempta based on transcriptome analysis. It seems that the analysis is ok, but there are several major problems which need to be solved carefully. If not, the MS cannot be considered for the next step of the submission. As follows:
1. In the work, authors identified 35 OBPs and 29 CSPs which contain intact open reading frames based on transcriptome analysis. I wonder that did the authors clone the 35 OBPs and 29 CSPs and sequence them by Sanger sequencing? I did not find the information of the primers, and I am not sure the authors conducted above things. From the perspective of myself, the clone and Sanger sequencing are essential for the identification. Please make it clear for me.
2. About the expression analysis of the OBP and CSP genes, in the MS, authors worked on the OBP and CSP genes of S. exempta, but, in the line 131 to 133, why “the expression of the identified OBPs and CSPs genes was calculated based on the TPM values from the transcriptome data of S. exempta, S. frugiperda, and S. littoralis”?
3. About the Sex- and stage-dependent expression patterns of OBP and CSP genes in the MS, the authors identified the differentially expressed genes (DEGs) by the TPM values, but after that, no further work of validation was performed. In my opinion, real-time quantitative PCR needs to be conducted and then, the DEGs can be validated. Why the authors neglected the validation?

Reviewer 2 ·

Basic reporting

Dong et al. investigated the OBPs and CSPs in a major global crop pest (Spodoptera exempta), sequence alignment indicated that 30 OBPs and 23 CSPs completely contained the conserved cysteines. Their data with whole bodies indicated that all highly expressed OBPs and CSPs in adults were male biased or were not different between the sexes, suggesting diverse functions of OBPs and CSPs in adult insects. However, there are several issues that must be addressed before it can be accepted for publication.

1. L32, L43, some contents are repetitive and need to be modified.
2. L75-77, the expression is not clear, please add relevant references to support this conclusion.
3. L82, add the two references: Zhang et al., 2013, plos one, 10.1371/journal.pone.0069715b; Zhang et al., 2017, peerj, 10.7717/peerj.3157.
4. L87, add the two references: Zhang et al., 2017, FIP, 10.3389/fphys.2018.00432, Li et al., 2021, BER, 10.1017/S0007485321000109.
5. Figure 1, Lack of signal peptide sequences. I suggest labeling OBP with complete ORF and different types of OBPs: classic, minus-c, plus-c…
6. Figure 2, I suggest labeling CSP with complete ORF.
7. Figure 3 and 4, the non full-length ORF genes should be deleted.
8. I suggest that the authors randomly select some genes for quantitative PCR to verify the data in Figure 8 and 9.

Experimental design

Dong et al. investigated the OBPs and CSPs in a major global crop pest (Spodoptera exempta), sequence alignment indicated that 30 OBPs and 23 CSPs completely contained the conserved cysteines.

Validity of the findings

Their data with whole bodies indicated that all highly expressed OBPs and CSPs in adults were male biased or were not different between the sexes, suggesting diverse functions of OBPs and CSPs in adult insects.

Additional comments

Dong et al. investigated the OBPs and CSPs in a major global crop pest (Spodoptera exempta), sequence alignment indicated that 30 OBPs and 23 CSPs completely contained the conserved cysteines. Their data with whole bodies indicated that all highly expressed OBPs and CSPs in adults were male biased or were not different between the sexes, suggesting diverse functions of OBPs and CSPs in adult insects. However, there are several issues that must be addressed before it can be accepted for publication.

1. L32, L43, some contents are repetitive and need to be modified.
2. L75-77, the expression is not clear, please add relevant references to support this conclusion.
3. L82, add the two references: Zhang et al., 2013, plos one, 10.1371/journal.pone.0069715b; Zhang et al., 2017, peerj, 10.7717/peerj.3157.
4. L87, add the two references: Zhang et al., 2017, FIP, 10.3389/fphys.2018.00432, Li et al., 2021, BER, 10.1017/S0007485321000109.
5. Figure 1, Lack of signal peptide sequences. I suggest labeling OBP with complete ORF and different types of OBPs: classic, minus-c, plus-c…
6. Figure 2, I suggest labeling CSP with complete ORF.
7. Figure 3 and 4, the non full-length ORF genes should be deleted.
8. I suggest that the authors randomly select some genes for quantitative PCR to verify the data in Figure 8 and 9.

Reviewer 3 ·

Basic reporting

The English language need to be improved by the native speakers or professional science editing service.

Experimental design

This is a gene sequence analysis and expression pattern comparison manuscript based on the previous transcriptome dataset without any new experiments conducted in this study. The analysis methods are suitable and well-rounded.

Validity of the findings

This study did the sequences analysis of S. exempta OBPs and CSPs based on previous transcriptome dataset and compare the expression patterns of them in different development stages. The draft may can provide the sequence info of SexeOPB/CSPs for further functional characterization of these proteins.

Additional comments

In this study, Dong et al., investigated the S. exempta OBP/CSPs sequence features and expression patterns in different development stages. 40 OBPs and 33 CSPs were identified based on the previous transcriptome dataset. The result provide a molecular basis for further functional characterization of these genes and also pest management control strategies. Please find the comments below which may help improve the draft.
Line 61-64: is this true? In my understanding, the OBPs function remain unclear. There are several models of the roles of OBPs in olfactory system. I would not like to say how OBP bind to odor molecules and help olfactory perception in sensillum lymph.
Line 91-92: change "fifth instar larvae on the first day of this stage larvae" to " the larvae on the first day of the fifth instar stage".
Line 127-128: change the sentence "...samples of S. exempta from single pairs, including the larvae on the first day of the fifth instar stage, pupae, and adult males and females,..." to "...samples of S. exempta from single gender mixed pairs, including the larvae on the first day of the fifth instar stage, pupae, and adult,..."
Line 135: it should be "one-way ANOVA" and at 0.05 significant level?
Line 164: change "thirty" to "30".
Line 169: change "twenty-three" to" 23".
Line 181-182: could you please add the principal component analysis method in the method section? and explain how you conduct the PCA.
Line 214: change "fourteen" to "14".

The transcriptome dataset access number should be provided somewhere in method section.

In the discussion section, could you add more deeper discussion of the function difference of OBP/CSPs among different species mentioned in the text? such as recent OBPs CRISPR KO studies. Can you make a table/figure like SI table3, with the info such as how many OBP/CSPs in different species mention in discussion and which group they belonged to?

The figure legends must be enriched. 1. For example, in figure 3/4 legend, please add what the x-axis showed? and how you get the Seqlogo (which software?) 2. in figure 5/6, what is other species in the tree? how many OBP genes in each species? It will be better provide more info in figure legend, not only in SI 1. 3. the insect name can be abbreviated in figure legends.

---

## Round 0.2 · Major Revisions

One reviewer still have comments and suggestions. Please revise the manuscript carefully according to them.

Reviewer 1 ·

Basic reporting

good

Experimental design

good

Validity of the findings

insufficient

Additional comments

In the revised MS, more results have been added and some of my problem have been solved. However, I still have some problems which confused me a lot.
In line 205-206, "For the OBPs,11 (SexeOBP9, 13, 15–16, 18, 26, 29–31, 35, and 37) of 40 were relatively highly expressed", and in the figure S2, the authors only showed nine of the OBPs, and why did the authors do that? Also, in line 225-226, "For the CSPs, 16 (SexeCSP2, 4–5, 8–10, 12–13, 17, 19, 23, 25–26, 28, 30, and 33) of 33 were relatively highly expressed", and in the figure S3, the authors only showed fifteen of the CSPs, and why?
Most importantly, what I mentioned in the last general comments letter about the validation is not just validating highly expressed OBPs and CSPs. What the authors need to do is to validate the results of DEGs which means that the authors should conduct qRT-PCR work on all the OBPs and CSPs from DEGs. Besides, in this work, as many published papers in this journal or others, the results of qRT-PCR validation is really important and reliable, in my opinion, the authors cannot simplify it.
One more question about the material of qRT-PCR samples, the authors did not describe how they collect the samples. Based on the descriptions in the MS, I am not sure that the authors used the samples from the original ones for the DGEs? Or the authors collected new ones for the qRT-PCR work. Please explain it clearly.

Reviewer 2 ·

Basic reporting

NA

Experimental design

I suggest that the author add details of insect rearing and tissue collection in the Materials & Methods section.

Validity of the findings

NA

Additional comments

I suggest that the author add details of insect rearing and tissue collection in the Materials & Methods section.

Reviewer 3 ·

Basic reporting

The authors have revised the manuscript according reviewers' comments. The revised version is clear and in good shape.

Experimental design

no comment

Validity of the findings

no comment

Additional comments

The authors have revised the manuscript according reviewers' comments. The revised version is clear and in good shape.

---

## Round 0.3 · accepted · Accept

The manuscript has undergone two rounds of major revisions. I have evaluated the revisions in light of the last decision round and I am satisfied that everything the Reviewers pointed out was addressed

Reviewer 1 ·

Basic reporting

no comment

Experimental design

no comment

Validity of the findings

no comment

Additional comments

no further comments